# The enpp4 ectonucleotidase regulates kidney patterning signalling networks in *Xenopus* embryos

Karine Massé [1,2,3 ✉], Surinder Bhamra[1], Christian Paroissin[4], Lilly Maneta-Peyret[5], Eric Boué-Grabot [2,3] &
Elizabeth A. Jones[1]

The enpp ectonucleotidases regulate lipidic and puringergic signalling pathways by controlling the extracellular concentrations of purines and bioactive lipids. Although both pathways are key regulators of kidney physiology and linked to human renal pathologies, their roles during nephrogenesis remain poorly understood. We previously showed that the pronephros was a major site of enpp expression and now demonstrate an unsuspected role for the conserved vertebrate enpp4 protein during kidney formation in *Xenopus*. Enpp4 over-expression results in ectopic renal tissues and, on rare occasion, complete mini-duplication of the entire kidney. Enpp4 is required and sufficient for pronephric markers expression and regulates the expression of RA, Notch and Wnt pathway members. Enpp4 is a membrane protein that binds, without hydrolyzing, phosphatidylserine and its effects are mediated by the receptor s1pr5, although not via the generation of S1P. Finally, we propose a novel and non-catalytic mechanism by which lipidic signalling regulates nephrogenesis.

[1] School of Life Sciences, Warwick University, Coventry CV47AL, UK. [2] Université de Bordeaux, Institut des Maladies Neurodégénératives, UMR 5293, F-33000 Bordeaux, France. [3] CNRS, Institut des Maladies Neurodégénératives, UMR 5293, F-33000 Bordeaux, France. [4] Université de Pau et des Pays de l'Adour, Laboratoire de Mathématiques et de leurs Applications—UMR CNRS 5142, 64013 Pau cedex, France. [5] Université de Bordeaux, CNRS, Laboratoire de Biogenèse Membranaire UMR 5200, F-33800 Villenave d'Ornon, France. ✉email: karine.masse@u-bordeaux.fr

Vertebrate kidney organogenesis is orchestrated by numerous signalling pathways and transcription factors regulating the proliferation and differentiation of diverse cell types to form the functional kidney. Despite the differences in complexity and organization of the three vertebrate kidneys, pronephros, mesonephros and metanephros, there is a remarkable conservation of molecular mechanisms during their development[1]. The pronephros, the functional embryonic kidney in amphibians, is a simple, easily accessible organ, which displays structural similarities to the other more complex kidney forms. Therefore, it has become an ideal model system to study molecular regulation during nephrogenesis and renal pathologies[2–5].

In mammals, lipidic and purinergic pathways regulate metanephric physiology and their deregulation has been linked to acute renal injury and chronic kidney diseases including renal fibrosis polycystic kidney disease, renal cell carcinoma, nephritis or diabetic nephropathy[6–10]. However, their potential roles during renal development have not been fully established, although the bioactive lipid sphingosine-1-phosphate (S1P) has been implicated during kidney branching[11]. Purines, mostly ATP and its derivatives, and bioactive lipids, S1P and lysophosphatidic acid (LPA), can function as extracellular ligands for G protein-coupled cell surface receptors[12,13]. Their availability for these receptors, in the extracellular space, is regulated by the activities of several membrane-bound enzymes, such as the ectonucleotidases, which are major regulators of renal health and disease[10,14]. The enpp (ectophosphodiesterase/nucleotide phosphohydrolase) proteins, which belong to the ectonucleotidase subfamily, are key regulators of both purinergic and lipidic signalling pathways with their dual enzymatic activities of hydrolysing purines and generating S1P and LPA bioactive lipids[15]. We have demonstrated that the pronephros is the major site of expression for the amphibian enpp genes family, in particular, enpp4 is highly expressed in *Xenopus laevis* pronephric tubules[16]. These data provided the first temporal and spatial embryonic expression profile for this evolutionarily conserved enzyme which remains functionally poorly understood[17–19]. In the present study, we investigated the function of enpp4 during pronephric development.

We demonstrate that Enpp4 function is crucial during kidney formation. While its knock-down leads to kidney formation defects, the overexpression of wild-type Enpp4, but not an inactive enzymatic protein, induces the formation of ectopic pronephroi characterized mostly by the presence of proximal tubule markers but in rare occasion of more distal tubule markers. These effects are mediated by the lipidic receptor S1pr5 and we also show that Enpp4 specifically binds to phosphatidylserine, implying a role for bioactive lipids in pronephrogenesis. Finally, we provide evidence that enpp4 misexpression alters the expression of members of the Notch, Wnt and RA signalling pathways and we propose a model for the mechanisms of action for Enpp4 and lipidic signalling in kidney development.

## Results

**Overexpression of Enpp4 results in ectopic pronephric tubules formation.** To analyse potential functional roles of Enpp4 during pronephros development, we first undertook a gain of function approach by performing immunostaining with pronephric tubules specific antibodies[20] on stage 41 embryos (Fig. 1a–o, and Supplementary Table 1 for raw data and statistical analyses). Enpp4 overexpression altered proximal pronephric tubules formation, in nearly 50% of the analysed embryos and induced ectopic (23%) and enlarged (18%) regions of the 3G8 staining domain ($n = 91$, Fig. 1b–e; Supplementary Table 1). Distal tubules were less affected, with 31% of the analysed embryos

displaying abnormal 4A6 staining. Ectopic 4A6 staining was rare (2%), with enlarged more distal tubule staining being the predominant phenotype (20%). Enpp4-induced phenotypes are significantly different compared to those of LacZ controls (3G8: $p < 0.001$ and 4A6: $p < 0.05$). Ectopic pronephroi were observed only when injections were performed into regions fated to become the lateral region of embryos (V2 blastomere) (Supplementary Fig. 1a). *Enpp4* overexpressing embryos displaying ectopic 3G8 staining ($n = 5$) were analysed by transverse section. Eight of the nine ectopic tubules sectioned had epithelial tubule structure complete with a lumen (Fig. 1f–j), which were similar to normal pronephric tubule structure (Fig. 1k, l). Similar significant renal phenotypes were observed following mouse *Enpp4* mRNA injection ($p < 0.001$, $n = 63$; Fig. 1m). However, no ectopic pronephric tubules were observed upon overexpression of Enpp4 mutant constructs containing a point mutation in the putative catalytic domain (T72A, T72S) or metal cation binding domain (D36N, D189N) (Fig. 1n, o)[16]. These data suggest that ectopic proximal pronephric tissues formation caused by Enpp4 overexpression depends on its catalytic activity.

**Overexpression of Enpp4 disturbs proximal-distal patterning of pronephros.** To further investigate this phenotype, embryos injected with *enpp4* mRNA were examined at stage 37 by whole-mount in situ hybridization using pronephric specific markers, *slc5a1.1*, *slc12a1*, *clcnkb* and *gata3*, which mark the various proximal/distal tubule segments[21,22] (Fig. 1a and p–s, see also Supplementary Table 1). Interestingly, Enpp4 overexpressing embryos showed ectopic and enlarged staining of *slc5a1.1* (proximal tubule marker, ectopic 30%, enlarged 14%; $n = 57$; $p < 0.001$) and *slc12a1* (marker of intermediate tubules, ectopic 17%, enlarged 25%; $n = 57$, $p < 0.001$) domains (Fig. 1p–q). Injection of *enpp4* mRNA failed to induce any separate ectopic *clcnkb* expression although the normal domain of expression (intermediate and distal tubule) was somewhat enlarged on the injected side (19%, $n = 58$, $p < 0.01$; Fig. 1r). The *gata3* expression domain (distal and collecting tubules) was relatively normal, although its anterior limit of expression, determined relative to the somite number, was slightly more posterior in more than half of the injected embryos (58%, $n = 43$, $p < 0.001$; Fig. 1s). This might reflect a change in anterior/posterior patterning induced by Enpp4 overexpression.

Injection of *enpp4* mRNA induced enlarged and reduced expression domains of both glomus marker *wt1* and *nphs1* at stage 33/34 but ectopic glomus staining was only observed in rare cases (Fig. 1t–u; Supplementary Table 1a). Although the statistical significance of these phenotypes was demonstrated (Supplementary Table 1b), we were not able to conclude on the exact Enpp4 effects on this structure.

Taken together, the results demonstrate that *enpp4* mRNA injection altered pronephros formation, leading to enlarged expression domains of markers of the entire tubule segments and to ectopic pronephric structures containing mostly domains of proximal and, in rare occasions, distal tubules marker genes.

**Overexpression of Enpp4 upregulates early kidney markers expression without altering mesoderm formation.** Embryos injected with *enpp4* mRNA were also examined by whole-mount in situ hybridization using early pronephros anlagen markers *irx1*, *lhx*1, *pax8*[23,24] and compared to *lacZ* mRNA injected embryos (see Supplementary Table 1). At stage 28, expression of both *lhx1* (61%, $n = 51$) and *pax8* (70%, $n = 50$) was significantly ($p < 0.001$) expanded especially in posterior parts of pronephric anlagen, with areas of intense staining consistent with that of the more anterior presumptive tubules (Fig. 1v–w). At early neurula

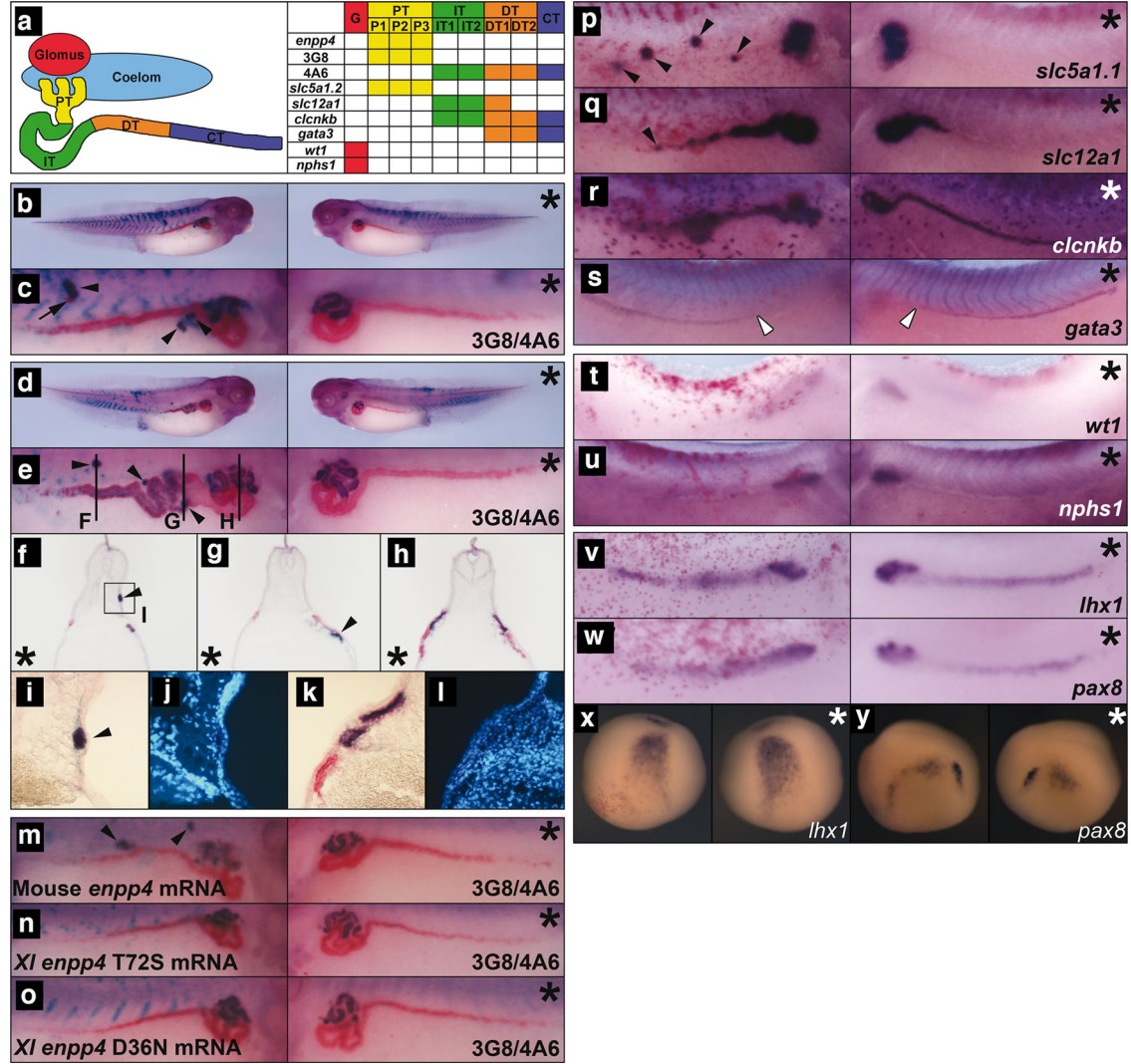

**Fig. 1 Overexpression of *enpp4* induces ectopic proximal pronephric tubules. a** Schematic diagram of pronephric structural components showing the expression domain for each marker used in this study, adapted from ref. [21]. G: glomus, PT: proximal tubule, IT: intermediate tubule, DT: distal tubule, CT: collecting tubule. **b–y** Embryos injected with 2 ng of *enpp4* and 250 pg of *LacZ* mRNAs were examined by 3G8/4A6 antibody staining (**b–o**) or whole-mount in situ hybridization with the following probes: *slc5a1.1* (**p**), *slc12a1* (**q**), *clcnkb* (**r**) and *gata3* (**s**) at stage 37/38; *wt1* (**t**) and *nphs1* (**u**) at stage 32; *lhx1* (**v, x**) and *pax8* (**w, y**) at stages 28 and 14. **f–l** Transverse sections of the embryo shown in panels (**d**) and (**e**) were cut in the anterior–posterior registers indicated by lines in panel (**e**). A higher magnification image (**i**) of ectopic pronephros in the somite indicated by square in (**f**) and of control kidney (**k**) and counterstained with Hoechst to indicate nuclei (**j, l**). Embryos injected with 2 ng of mouse wild-type *Enpp4* (**m**), *X. laevis* mutated in the putative catalytic site (**n**) or in the cation binding site (**o**) and 250 pg of *LacZ* mRNAs were examined by 3G8/4A6 antibody staining. The asterisk denotes the uninjected side of each embryo. Arrowheads indicate ectopic marker staining. Blank arrowheads in (**s**) indicate the anterior limit of *gata3* expression. See also Supplementary Table 1 for raw data and statistical analyses and Supplementary Fig. 1.

stages (Fig. 1x–y), the expansion of *pax8* (17%, *n* = 70) expression domain was also observed (Fig. 1y). Furthermore, ectopic *pax8* expression was also induced following *enpp4* RNA injection at both stages analysed (neurula stage, 21%; stage 28, 2%; *p* < 0.001; Supplementary Table 1b). *Lhx1* expression was also altered at early neural stages, but no ectopic *lhx1* expression was observed (Fig. 1x, Supplementary Table 1b). *Furthermore, irx1* expression domain was not altered following Enpp4 overexpression (Supplementary Fig. 1b).

Since normal somite development is a prerequisite for pronephros development, *enpp4* RNA injected embryos were analysed by whole-mount in situ hybridization using the muscle marker *myh4* at stage 33/34. The expression pattern was normal in all injected embryos (*n* = 55; Supplementary Fig. 1c; Supplementary Table 1). Enpp4 overexpression did not also alter the expression of pan-mesoderm marker, *xbra* at stage 10.5 (*n* = 46;

Supplementary Fig. 1d and Supplementary Table 1). Therefore, we conclude that, *enpp4* mRNA injection had no gross effects on mesoderm induction per se or on somite development.

**Morpholino knock-down of *enpp4* results in smaller pronephros formation**. To determine whether Enpp4 is required for normal pronephros development, we have undertaken a loss of function approach using two specific anti-sense morpholino oligonucleotides (MOs) (Fig. 2, Supplementary Fig. 2a–c for specificity and efficiency of the MOs and Supplementary Table 2). The overall morphology of the embryos appeared normal in *enpp4* morphants. Moreover, the expression pattern of the somitic *myh4*, muscle *myoD* and early mesoderm *xbra* markers was unaffected in *enpp4* MO1 injected embryos (*myh4* 100%, *n* = 47 and *xbra* 84%, *n* = 19; Supplementary Fig. 2d, e) suggesting that

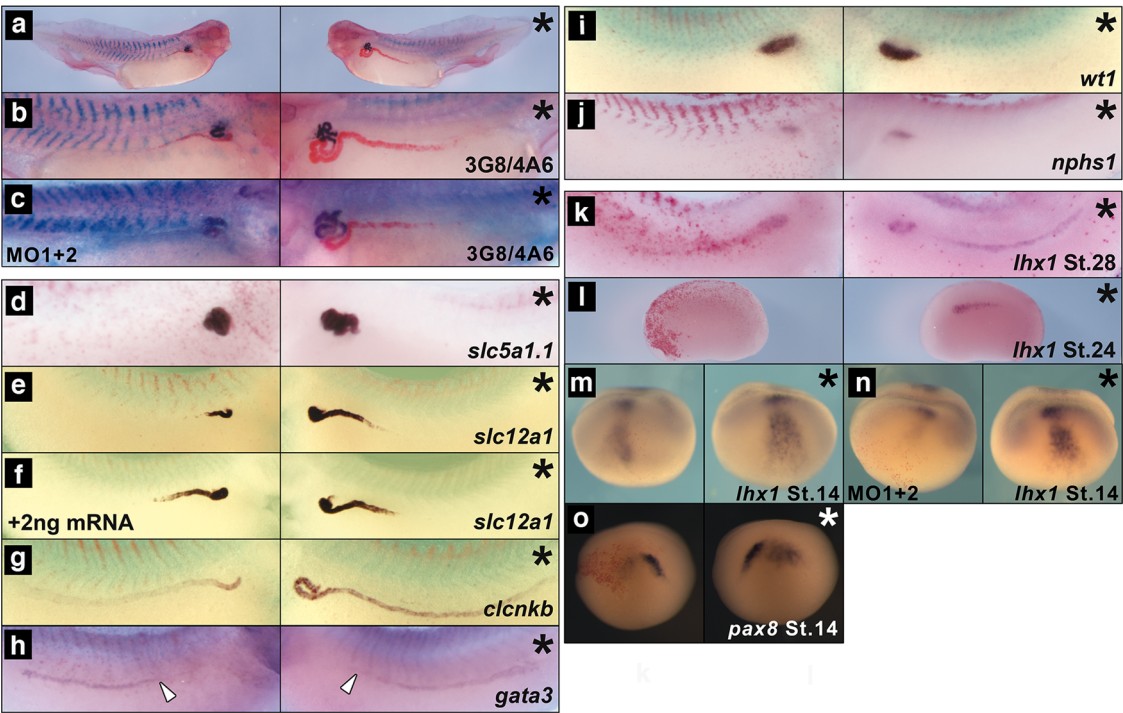

**Fig. 2 MO knock-down of *enpp4* expression disrupts pronephros formation.** Embryos targeted with 10 ng of *enpp4* MO1 or 10 ng of both *enpp4* MOs (**c**, **n**) and 250 pg of *lacZ* mRNA were examined by 3G8/4A6 antibody staining (**a**–**c**) or whole-mount in situ hybridization with the following probes: *slc5a1.1* (**d**), *slc12a1* (**e**, **f**), *clcnkb* (**g**) and *gata3* (**h**) at stage 37/38; *wt1* (**i**) and *nphs1* (**j**) at stage 33/34, *lhx1* (**k**–**n**) at stages 28, 24 and 14, *pax8* (**o**) at stage 14. The embryo shown in (**f**) was co-injected with 2 ng of mouse *Enpp4* mRNA to rescue enpp4 knock-down phenotype. The asterisk denotes the control, uninjected side of each embryo. Blank arrowheads in (**h**) indicate the anterior limit of *gata3* expression. See also Supplementary Table 2 for raw data and statistical analyses and Supplementary Figs. 2 and 3.

any kidney phenotypes observed in *enpp4* morphants are not due to general mesoderm defects.

Injection of *enpp4* MO1 resulted in a significant reduction of expression of both 3G8 (65%, $p < 0.001$) and 4A6 (28%, $p < 0.05$) ($n = 107$, Fig. 2a, b, Supplementary Table 2), indicating that a smaller pronephros had formed. A similar phenotype was observed following *enpp4* MO2 injection (3G8 49%, 4A6 24%, $n = 87$; Supplementary Fig. 2f, g) and was worsened when both MO were injected together (Fig. 2c). These results indicate that Enpp4 is required for both proximal and distal pronephric tubule development.

Enpp4 knock-down in embryos showed a significant reduced expression of *slc5a1.1* (58%, $n = 64$, $p < 0.001$) and *slc12a1* (56%, $n = 75$, $p < 0.01$) with MO1 (Fig. 2d, e) as well as with MO2 (Supplementary Fig. 2h, i). Rescue experiments performed by co-injecting mouse *Enpp4* mRNA (2 ng) with *enpp4* MO1 or MO2 (10 ng each, $n = 72$ or 28, respectively) restore partially but significantly ($p < 0.001$) the normal phenotype of *slc12a1* staining domain confirming the specificity of the knock-down of *enpp4* expression on pronephric development (Fig. 2f and Supplementary Fig. 2j). Ectopic *slc12a1* expression was also observed in some embryos (11% and 50% with MO1 or MO2, respectively) consistent with Enpp4 overexpression phenotype. The *clcnkb* expression in the intermediate tubules and *gata3* anterior expression domain were reduced after *enpp4* MO1 (53.5%, $n = 40$ and 31%, $n = 42$, respectively) but these differences are not statistically significant $p > 0.05$) (Fig. 2g, h). Enpp4 knock-down has no effect on glomus formation, as the expression of *wt1* and *nphs1* was normal in most of *enpp4* MO1 injected embryos at stage 33/34 (wt1 = 94%, $n = 34$ and nphs1 = 83%, $n = 41$) (Fig. 2i, j). These results suggest that *enpp4* knock-down affected pronephric

tubule, especially proximal and intermediate segments, differentiation rather than just the proximal-distal patterning of pronephric tubule segmentation.

To address potential Enpp4 roles during early phases of pronephros development, we tested by RT-PCR its expression at key stages during kidney development in dissected developing pronephric tissues (Fig. 3a). At later stages, *enpp4* expression profile is in agreement with our published in situ hybridization data[16]. However, weak expression is also detected in the embryonic kidney from stage 12.5 and is upregulated by stage 26. These data suggest that Enpp4 might be involved during early pronephric developmental phases. The expression domain of *lhx1* and *pax8* was altered following knock-down of *enpp4* by MO injection (see Supplementary Table 2 for raw data and statistical analyses). At stage 28, expression of *lhx1* was reduced especially in posterior elements of the pronephric anlagen (35%, $n = 23$; Fig. 2k), although its expression in presumptive proximal tubules was sometimes unaffected or expanded. At stage 24, the expression domain of *lhx1* was clearly reduced (75%, $n = 20$, Fig. 2l), suggesting involvement of Enpp4 in early pronephros differentiation. Injection of *enpp4* MO2 caused similar phenotypes at both stages (stage 28, 30%, $n = 20$; stage 24, 55%, $n = 20$; Supplementary Fig. 2k–l). At early neurula stages, expression of *lhx1* was also reduced following *enpp4* MO1 or MO2 injection (MO1 37%, $n = 46$; MO2 64%, $n = 61$; $p < 0.001$; see also Supplementary Table 2) and even absent after MO1 injection (MO1 31%, $n = 46$) (Fig. 2m and Supplementary Fig. 2m). The injection of both MO's resulted in a stronger reduction of *lhx1* expression ($n = 31$; Fig. 2n). A reduction of *pax8* expression was also observed at the neurula stages following *enpp4* MO1 or MO2 injection (Fig. 2o and Supplementary Fig. 2n).

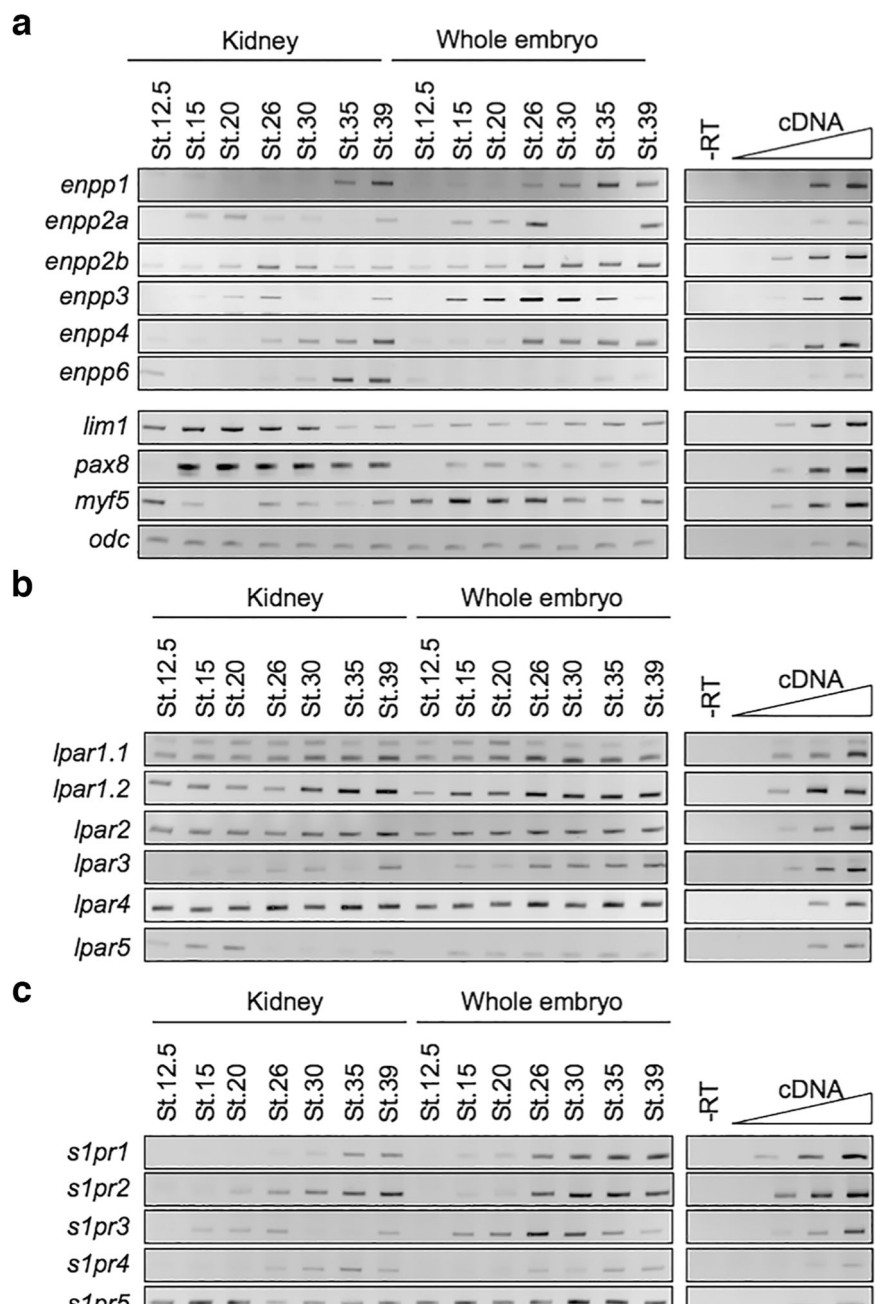

**Fig. 3 The *enpp* and the lipidic receptors, the *lpar* and *s1pr*, gene family members are expressed in the pronephros.** Developing pronephric anlagen or pronephric tubules were dissected as indicated, from whole *X. laevis* embryos and total RNA extracted. RT-PCR was performed on pronephric dissected tissues and control whole embryos along with negative and linearity controls. **a** Comparative expression pattern of the *enpp* genes and pronephric and muscle marker genes controlling the quality of the dissections. **b** Comparative expression profile of the *lpa* receptors. **c** Comparative expression profile of the *s1pr* genes.

Taken together, the Enpp4 knock-down and rescue experiments demonstrate that normal levels of *enpp4* expression are required for normal pronephric development.

***Enpp4* misexpression phenotypes are distinct from those following *enpp6* misexpression**. To address if the ectonucleotidase Enpp6, also expressed in the proximal pronephric tubules[16], can compensate for Enpp4 loss of function, we performed single or double *enpp4/enpp6* knock-down and rescue experiments (Supplementary Fig. 3 and Supplementary Table 2). Enpp6 depletion induced the formation of a smaller pronephros on the injected side, in the similar frequency than *enpp4* knock-down

(Supplementary Fig. 3b, c). The co-injection of *enpp4* MO2 and *enpp6* MO resulted in the formation of reduced 3G8 (64%, $n = 76$) and 4A6 (62%, $n = 76$) positive tissues and was not statistically different from the effects of *enpp4* MO2, demonstrating that the *enpp6* MO did not worsen the renal phenotype caused by *enpp4* knock-down (Supplementary Fig. 3a). Pronephric formation was altered following Enpp6 overexpression, with reduced or absent, but never ectopic, pronephros observed and injection of *enpp6* mRNA did not rescue the *enpp4* MO2 phenotype (Supplementary Fig. 3d, e). Taken together, these data suggest that there is no functional redundancy between Enpp4 and Enpp6 ectonucleotidases.

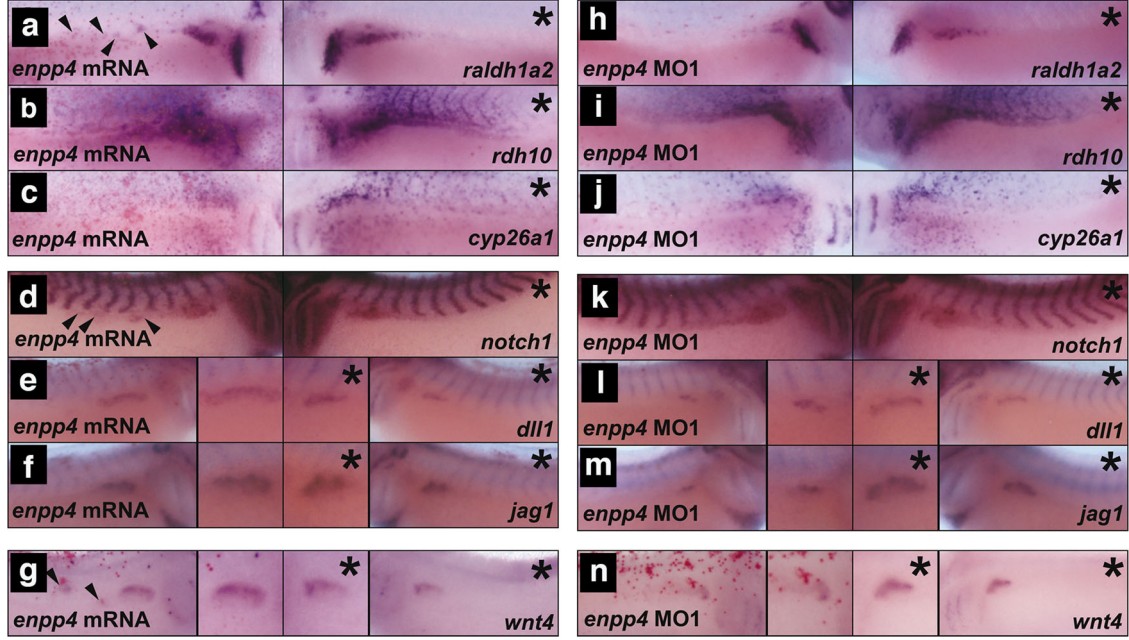

**Fig. 4 Microinjection of *enpp4* mRNA and MO affect the expression of retinoic acid synthesis enzymes, notch and wnt signalling molecules. a–g** Embryos targeted with 2 ng of *enpp4* and 250 pg of *LacZ* mRNA or **h–n** 10 ng of *enpp4* MO1 and 250 pg of *LacZ* mRNA were fixed at stage 28 and examined by whole-mount in situ hybridization with the following probes: *raldh1a2* (**a, h**), *rdh10* (**b, i**), *cyp26a1* (**c, j**), *notch1* (**d, k**), *dll1* (**e, l**), *jag1* (**f, m**) and *wnt4* (**g, n**). The asterisks denote the control, uninjected sides. Arrowheads indicate ectopic staining of the marker gene (see also Supplementary Table 3 and Supplementary Fig. 4).

***Enpp4* misexpression affects expression of several components of the RA, Notch and Wnt signalling pathways.** Since retinoic acid (RA), Notch and Wnt signalling pathways are involved in pronephros formation and patterning[25–30] and the timing of the endogenous expression of many components of these pathways overlap, we hypothesized that Enpp4 might affect these pathways. We therefore examined embryos injected with *enpp4* mRNA (Fig. 4a–g) or *enpp4* MO1 (Fig. 4h–n) by in situ hybridization for alterations in expression domain of representative members e.g. ligands (Dll1, Jag1, Wnt4) receptors (Notch1) and metabolic enzymes (Raldh1a2, Rdh10, Cyp26a1) of these three pathways (see also Supplementary Table 3).

Enpp4 overexpression induced ectopic and enlarged *raldh1a2* and *rdh10* expression domains in the pronephric region ($p < 0.001$, $n = 37$ and 34, respectively, Fig. 4a, b and Supplementary Table 3). *Enpp4* knock-down reduced their expression in pronephric region of ~20% of analysed embryos but this phenotype is not significant ($p > 0.05$, $n = 33$ for each probe, Fig. 4h, i and Supplementary Table 3). *Raldh1a2* expression was unaffected in the pharyngeal arches. In contrast, *cyp26a1* expression was normal in the pronephric region following *enpp4* mRNA or MO injection (98%, $n = 44$ and 88%, $n = 42$, $p > 0.05$, respectively) (Fig. 4c, j). Both *enpp4* mRNA and MO injection disturbed *rdh10* and *cyp26a1* expression in the somites. These results suggest Enpp4 controls the expression of enzymes involved in RA synthesis and potentially might act upstream of RA signalling. To verify this, *enpp4* expression, along with *pax8* and *lhx1*, was analysed in animal caps. *Enpp4* expression is not induced in animal caps treated with RA for 3 h (Supplementary Fig. 4). This confirms the epistatic relationship between Enpp4 and RA signalling.

*Enpp4* overexpression revealed significant enlarged expression domains of *notch1* (53% $n = 75$, $p < 0.001$, Fig. 4d). *Notch1* expression was normal in the majority of embryos after *enpp4* knock-down (80%, $n = 41$, Fig. 4k). *Enpp4* mRNA injection caused ectopic (40%) and enlarged (23%) expression domains of

*dll1* ($n = 81$, $p < 0.001$ Fig. 4e), while MO-injected embryos showed *dll1* reduced expression (33%, $n = 46$, Fig. 4l). Ectopic (20%) and enlarged (44%) *jag1* expression domains were observed following Enpp4 overexpression ($n = 81$, $p < 0.001$, Fig. 4f), while MO1 injection reduced its expression domain (in 38% of the analysed embryos, $n = 42$, Fig. 4m). These results suggest that Enpp4 also regulates members of the Notch signalling pathway and that *jag1* expression is more affected by *enpp4* depletion than *dll1* expression. Since Rnfg overexpression caused ectopic pronephroi formation[28], we further addressed the link between Enpp4 and the Notch pathway by injecting *enpp4* mRNA or MO2 in presence of *rfng* mRNA or MO. Our data show that modulation of notch-ligand interactions by Fringe proteins alters Enpp4 pronephric phenotypes, although differences are not significant (Supplementary Fig. 5 and Supplementary Table 3).

Finally, *enpp4* mRNA injected embryos showed enlarged (32%) and ectopic (17%) *wnt4* expression domains ($n = 41$, Fig. 4g) while expression of *wnt4* was reduced in most of the *enpp4* MO1 injected embryos (82%, $n = 39$, $p < 0.001$ Fig. 4n, Supplementary Table 3). These data suggest that Enpp4 is necessary and sufficient for promoting pronephric *wnt4* expression.

***Xenopus* Enpp4 is localized to the plasma membrane.** To address the cellular localization of the amphibian Enpp4 protein, we generated a specific polyclonal antibody against the full-length *Xenopus* protein (see Supplementary Table 4 for specificity evaluation of the antibody) and expressed *Xenopus* wild type (WT), T72S mutant and mouse Enpp4 cDNA in CHO cells by transient transfection. *Xenopus* Enpp4 WT protein was detected, by western blotting, in whole cells and in the membrane fractions, but not in the soluble fractions (Fig. 5a). Immunofluorescence experiments confirmed Enpp4 expression at the cell membrane (Fig. 5b). These results show that *Xenopus* Enpp4 is a trans-membrane protein, as its mouse ortholog[31]. Unfortunately, we failed to detect the endogenous Enpp4 expression in *Xenopus* embryos using the anti-XlEnpp4 antibody.

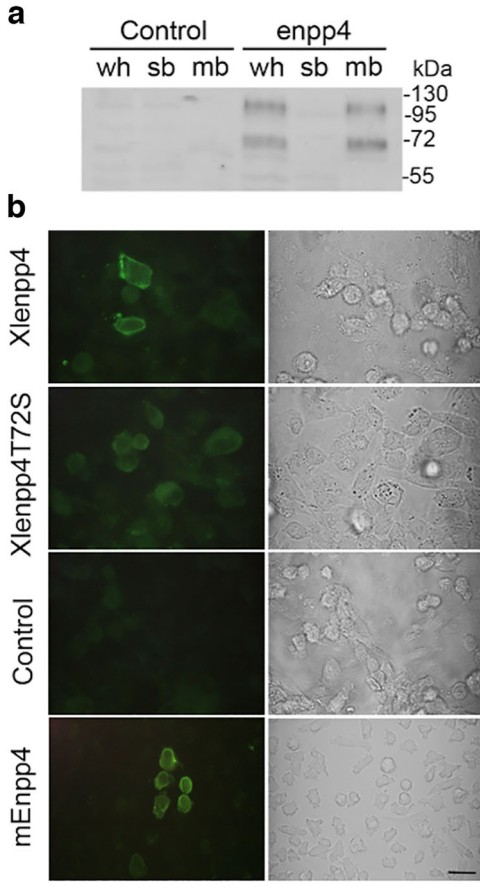

**Fig. 5 Enpp4 is a transmembrane ectonucleotidase. a** CHO cells were transfected with Xl*enpp4*-pcDNA3.1 or empty vector (control) and the cellular distribution of Enpp4 determined by western blotting using anti-XlEnpp4 antibody and proteins extracts from membrane (mb), soluble (sb) or whole cells (wh) fractions. **b** Representative images of the cellular distribution of Enpp4 determined by immunofluorescence using anti-XlEnpp4 or anti-mEnpp4 antibodies from CHO cells transfected with Xl*enpp4*-pcDNA3.1, Xl*enpp4T72S*-pcDNA3.1, *mEnpp4*-pcDNA3.1 or empty pcDNA3.1 vector (control). Corresponding brightfield images are also presented. Scale bar represents 20 μm.

**Phospholipid receptors are expressed in the developing pronephros along with the *enpp4* gene.** Based on sequence homology of *Xenopus* Enpp family[16], Enpp4 is more related to lipid-hydrolysing Enpp6 and 7 enzymes. We therefore hypothesized that the roles of transmembrane-bound Enpp4 during pronephrogenesis might be linked to the lipidic signalling pathway. To test if phospholipid receptors might mediate Enpp4 functions, we established the expression profiles of *lpa* and *s1p* receptor family members previously identified[32] by RT-PCR in kidney dissected tissues (Fig. 3b, c). All *lpa* receptors, except *lpar3* and 5, are expressed in pronephric tissues at a similar level from the time of kidney specification to late differentiation, confirming their ubiquitous expression profile during *Xenopus* embryogenesis[32]. The *s1p* receptors display different expression profiles, with *s1pr5* being the only family member to be expressed in the developing kidney at every stage analysed, particularly in the presumptive pronephric tissue at stage 12.5. No such renal expression was detected by in situ hybridization in our previous study, although expression in marginal zone of blastula embryos was detected by RT-PCR[32]. These data suggest that the pronephric level of expression of these lipidic receptors, especially

s1pr5, is relatively low, under the in situ hybridization detection level.

**Overexpression of *s1pr5* enhances Enpp4 function to induce ectopic pronephros.** In order to identify whether a lipidic receptor is involved in Enpp4 phenotypes, *s1pr* and *lpar* overexpression analyses were carried out by injecting 2 ng of *s1pr5*, *s1pr1*, *lpa1.1* and *p2y10* mRNAs alone or in combination with 1 ng of *enpp4* mRNA alone (Fig. 6a–d; Supplementary Fig. 6a–f; Supplementary Table 5) i.e half of the optimal dose to generate ectopic pronephros, see Fig.1). At 1 ng *enpp4* mRNA dose ectopic 3G8 (7%) and 4A6 (5%) staining were obtained only in rare cases (n = 94, Fig. 6d) compared to the optimized dose of *enpp4* mRNA used in Fig. 1. Injection of any tested lipidic receptor mRNA alone does not induce any ectopic kidney formation with normal 3G8 and 4A6 staining in the majority of the embryos (n = 52, Fig. 6c, Fig. S6b, d, f). Only co-expression of *s1pr5* and *enpp4* mRNAs resulted significantly in higher ectopic 3G8 staining compared to *enpp4* mRNA alone (38%, n = 89, p < 0.001, Fig. 6a, b, Supplementary Fig. 6a, c, e and Supplementary Table 5). Furthermore, the size and frequency of the ectopic pronephroi were higher than with injections of 2 ng of *enpp4* alone (see Fig. 1b–e). These results indicate that, among the tested receptors, only S1pr5 enhanced Enpp4 function to generate ectopic pronephros.

**The functions of Enpp4 are mediated by the lipidic S1pr5 receptor.** To further confirm that S1pr5 is involved in pronephros development we performed loss of function experiments. Two *s1pr5* genes are identified in *X.laevis* genome and *s1pr5.L* corresponds to our published sequence[32]. Despite distinct spatial expression in the adult frog, the two *s1pr5* homeologs display a very similar expression profile during *X.laevis* embryogenesis and are both expressed in the pronephric tissues (Supplementary Fig. 7a, b). We therefore performed loss of function analyses with an anti-sense MO against *Xenopus s1pr5.L* and *s1pr5.S* (see Supplementary Figs. 6g, h and 7c, d for MOs efficiency and specificity evaluation and Supplementary Table 5 for raw data and statistical analyses). Embryos injected with 15 ng of *s1pr5.L* MO or *s1pr5.S* MO displayed significant reduced 3G8 and 4A6 staining (n = 43 and 66, respectively, Fig. 6e, f, Supplementary Fig. 7e and Supplementary Table 5) suggesting that S1pr5 receptors are required for normal pronephros formation in *Xenopus*.

To examine potential synergistic effects, we co-injected 7.5 ng of *s1pr5.L* MO with 5 ng of *enpp4* MO (half of the dose used previously for single injections) and compared their phenotype to those obtained following co-injection of *s1pr5* MO or *enpp4* MO with control MO (Fig. 6g–j). As expected, embryos co-injected with *s1pr5.L* and *enpp4* MOs generated the strongest phenotype and smallest pronephros, with strong reduction of 3G8 (74%) and 4A6 (81%) staining domains (n = 42, Fig. 6g, h). *Enpp4* MO alone also caused strongly reduced 3G8 (65%) and 4A6 (60%) staining as previously shown (n = 40, Fig. 6j), while the *s1pr5.L* MO alone reduced pronephric size in both 3G8 (22%) and 4A6 (29%) domains less frequently (n = 51, Fig. 6l). Although there are no significant differences in pronephric phenotype between *enpp4* MO1 + *s1pr5.L* MOs and *enpp4* MO1 injected embryos, we concluded that co-injection of *s1pr5.L* and *enpp4* MOs showed additive effects on the inhibition of *Xenopus* pronephros development based on the size of the scored pronephroi.

To further analyse the link between Enpp4 and S1pr5, we carried out injection of 2 ng of *enpp4* mRNA together with 15 ng of *s1pr5.L* MO (Fig. 6k–l) or control MO. As expected, injection of *s1pr5.L* MO lowered the percentage of embryos displaying

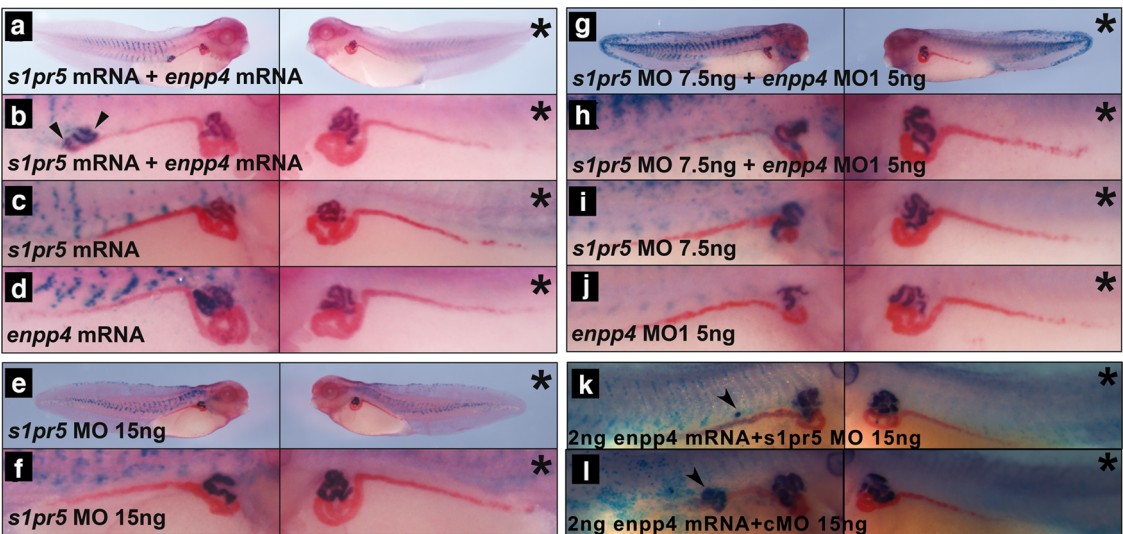

**Fig. 6 Enpp4 pronephric functions are mediated by the S1pr5 receptor.** Injected embryos were examined by 3G8/4A6 antibody staining following **a**, **b** double targeted injection of 2 ng s1pr5.l mRNA and 1 ng of enpp4 mRNA, **c** single targeted injection of s1pr5.l, **d** enpp4 mRNA or **e**, **f** embryos injected with 15 ng of s1pr5.L MO. **g**, **h** Double targeted injection of 7.5 ng of s1pr5.L MO and 5 ng of enpp4 MO1. **i**, **j** Single targeted injection of s1pr5.L MO (**i**) or enpp4 MO1 (**j**). **k**, **l** Double targeted injection of 15 ng of s1pr5.L MO and 2 ng of enpp4 mRNA (**k**) and 15 ng of cMO and 2 ng of enpp4 mRNA (**l**). An asterisk denotes the control uninjected side. An arrowhead indicates ectopic 3G8 staining (see also Supplementary Table 5 and Supplementary Figs. 6 and 7).

ectopic 3G8 and 4A6 staining caused by enpp4 mRNA injection (17.5% and 0%, respectively, $n = 40$, Fig. 6k; compared to 48% and 10%, $n = 50$, Fig. 6l). Furthermore, the size and number per embryos of these ectopic pronephroi was lower than with injection of 2 ng of enpp4 and 15 ng of cMO. Moreover, the percentage of embryos injected with enpp4 mRNA and s1pr5.L MO displaying a reduced 3G8 and 4A6 expression domain remains high (42.5% and 30%, respectively, $n = 40$, Fig. 6k), most certainly due to the loss of function of S1pr5.

These results indicate that the ectopic pronephric tissues induced by Enpp4 overexpression are due to the activation of the S1pr5 receptor.

**Enpp4 specifically interacts with the lysophospholipid phosphatidylserine.** To assess if Enpp4 hydrolyses lipids and generates a ligand, which could bind to the S1pr5 receptor, phospholipid binding was tested by a protein-lipid overlay assay using commercial pre-spotted lipid membranes. Out of the 26 bioactive lipids tested, only phosphatidylserine (PS) is specifically bound by Xenopus Enpp4 (Fig. 7a, b and Supplementary Fig. 8). Moreover, this interaction is abolished when the putative catalytic site is mutated (Fig. 7c). We then tried to determine the enzymatic activity of Xenopus Enpp4. However, we could not detect any lipid derivatives, e.g. DAG, PA or LysoPS, which could be generated from the hydrolysis of PS in the membrane proteins fractions from overexpressing Enpp4 CHO cells. Taken together, these data show Enpp4 specifically interacts with PS but does not have PLA, PLC or PLD activity towards PS.

## Discussion

This paper reports newly identified and unexpected roles of the conserved ectonucleotidase Enpp4 during vertebrate kidney development. Moreover, our findings provide a novel molecular mechanistic understanding for pronephric development and emphasizes the importance of the lipidic pathways in kidney formation (Fig. 8).

We previously showed that enpp4 is expressed in pronephric tubules[16], but our present data demonstrate that low but significant levels of enpp4 can be detected at the time of proximal tubule specification[33]. Weak enpp4 expression was previously

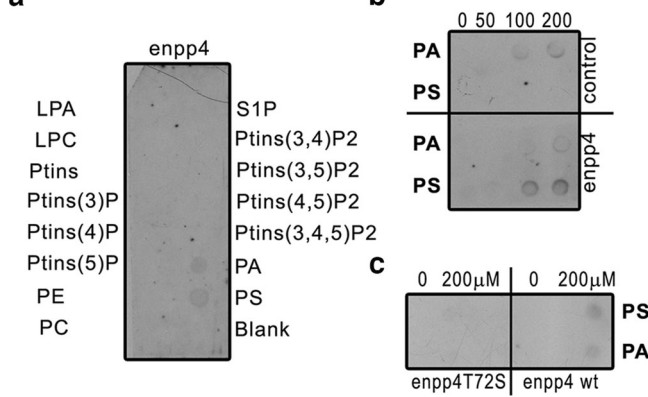

**Fig. 7 Enpp4 specifically binds to the lysophospholipid, phosphatidylserine. a** Membrane Lipid strip™ was incubated with membrane protein extracts from Enpp4 overexpressing CHO cells and the bound Enpp4 protein detected with anti-XlEnpp4 serum. **b**, **c** Nitrocellulose membranes were spotted with increasing amount of PA or PS and incubated with membrane protein extracts from CHO cells transfected with enpp4-pcDNA3.1, with enpp4T72S-pcDNA3.1 or empty plasmid (control) and the bound proteins detected with anti-XlEnpp4 serum. LPA lysophosphatidic acid, LPC lysophosphocholine, PA phosphatidic acid, PC phosphatidylcholine, PE phosphatidylethanolamine, PS phosphatidylserine, PtIns phosphatidylinositol, S1P shingosine-1-phosphate (see also Supplementary Fig. 8).

detected by RT-PCR but not by ISH, in gastrula embryo[16]. This discrepancy is attributable to the lower sensitivity of ISH compared to RT-PCR for the detection of gene expression patterns.

Our work demonstrates that Enpp4 regulates the expression level of two of the transcription factors involved in pronephric anlagen formation, lhx1 and pax8[23] but not irx1[24]. However, induction of lhx1 ectopic expression is delayed compared to pax8 one. Such a distinct expression regulation of these two pronephric genes by several signalling pathways has already been described[34,35]. As Lhx1 is necessary for the early patterning of the

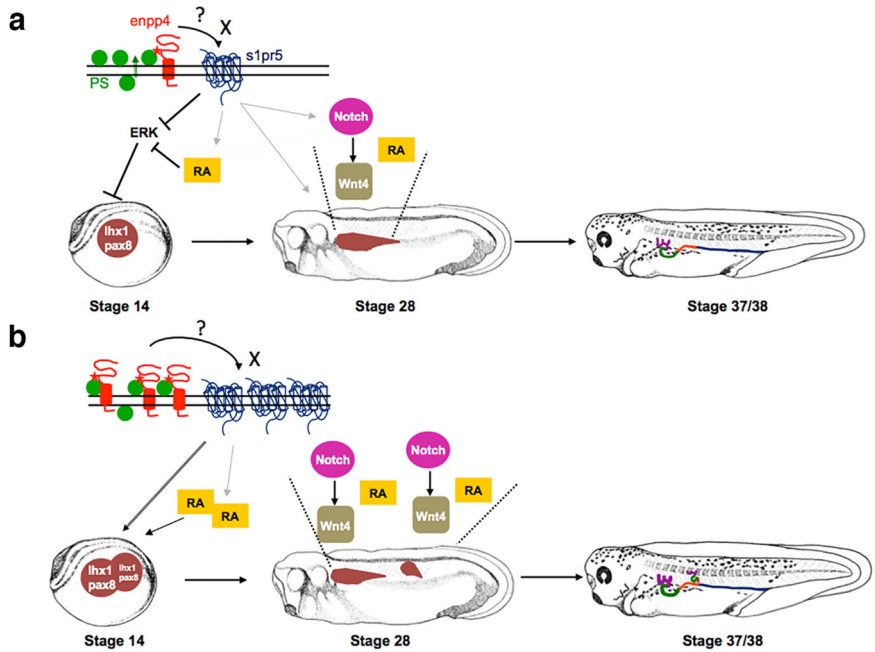

**Fig. 8 Proposed model of how Enpp4/S1pr5 controls pronephros patterning. a** During normal pronephric development, in the extracellular space, Enpp4 binds to phosphatidylserine close to, or in its catalytic site, which can then either interact with the S1pr5 or produce a novel ligand X, able to bind to this receptor. The activation of S1pr5 leads to the upregulation of *lhx1/pax8* pronephric markers in the kidney field either by acting upstream of RA signalling pathway or by acting directly *via* the ERK or calcium pathways. At later stages, RA is required for tubules morphogenesis and Notch and Wnt pathway are involved in the patterning of the pronephric tubules. The mechanism by which S1pr5 activation directs the expression domains of these genes remains to be confirmed. **b** Enpp4 and S1pr5 overexpression lead to expanded and ectopic expression domains for both the Notch and RA pathway genes and *wnt4*. These changes in patterning gene expression domains induce the formation of enlarged pronephric segments and ectopic pronephric tubules.

entire kidney and subsequently growth and elongation in the development of the pronephric tubules[36,37], the reduction of *lhx1* expression can explain the formation of the small pronephros in *enpp4* morphants. Pax8 is necessary for the earliest steps of pronephric development and for pronephric precursors cell proliferation and can induce the formation of ectopic pronephric tubules[23,38]. Therefore, the ectopic expression of *pax8* in Enpp4 overexpressing embryos can explain the formation of ectopic pronephroi. We also demonstrate that Enpp4 is sufficient to generate kidney, but only from lateral mesoderm and not in ectopic non-lateral positions. This suggests that the lateral mesoderm must contain either the receptor or the substrate necessary for Enpp4 function. Furthermore, the induction of *enpp4* expression in activin treated animal caps confirms the importance of mesoderm tissues for Enpp4 pronephric functions (see Supplementary Fig. 4).

The ectopic kidneys formed from *enpp4* overexpression consist of tubular structures and are patterned along their proximal/distal axis. Moreover, in some rare cases, there is a complete mini-duplication of the entire pronephros. This surprising phenotype could be explained by the upregulation of the patterning of signalling pathway members. RA signalling is required during gastrulation for pronephric specification. Increased levels of RA signalling by Enpp4 overexpression could lead to *pax8* expression activation and then to the formation of ectopic pronephroi[23,25]. RA signalling is also required post-gastrulation for tubules morphogenesis and its downregulation in *enpp4* morphants could explain tubules formation defects[25]. Subsequently, RA increase could pattern these ectopic tubules, as it has been shown during zebrafish pronephric nephron segmentation patterning[39,40]. However, in *Xenopus* pronephros, RA signalling increases expression level of distal tubules markers[41]. Moreover, RA signalling also regulates the expression of members of the Notch pathway[42], which can subsequently activate *wnt4* expression[28],

which then functions to pattern the proximal pronephros. Our data demonstrate that Rnfg protein is involved in mediating Enpp4 signalling, probably by its ability to modify Notch-ligand interactions[43]. Therefore, we speculate that Enpp4 acts upstream or in parallel to RA signalling and upstream of Notch and Wnt pathways (see Fig. 8). As *enpp4* expression was unchanged in animal caps treated with RA compared to control caps, this supports the hypothesis that Enpp4 acts upstream of the RA pathway.

A key question is how the misexpression of Enpp4, an ecto-nucleotidase, can alter gene expression. Phosphatidylserine translocation across the cell membrane is a well-known indicator of apoptosis but is also involved in physiological and developmental processes[44,45]. Therefore, Enpp4 could bind to PS in the extracellular space during pronephrogenesis. Enpp4 enzymatic activity is essential for ectopic kidney formation, suggesting that renal alterations are due to an excess or shortage of Enpp4 generated products in the extracellular space inducing cell responses *via* the activation of the S1pr5 receptor. However, our data strongly suggest that the Enpp4 kidney phenotype is not linked to the bioactive lipids LPA or S1P. The fact that the observed kidney phenotype might be due to a non-catalytic effect of Enpp4 might be puzzling and unexpected, especially since Enpp6 has been suggested to play major renal physiological role through its enzymatic functions[46]. However, specific functions of other Enpps, such as Enpp1, Enpp2 and Enpp5, have been shown to be independent of their enzymatic activity[47–50]. It is therefore possible that Enpp4 does not hydrolyse PS but its interaction with PS is necessary for the activation of Enpp4 and subsequently of S1pr5. PS binding and conformational change mechanisms have been demonstrated for protein kinase C activation in mammalian kidney cells, supporting this hypothesis[51]. Although we were unable to detect any of the predicted products of PS hydrolysis, we cannot rule out that we failed to characterize Enpp4 enzymatic

activity and that Enpp4 will generate a bioactive lipid, other than S1P, which is able to bind to S1p5r, the most divergent member of the S1pr family[32]. Although signalling through S1PR5 has been poorly studied, the activation of the S1P5 receptor has been linked to an intracellular calcium increase and inactivation of the ERK pathway, both pathways regulating pronephric field formation *via* RA signalling[35,52–55].

We show that mouse Enpp4 can fulfil *Xenopus* Enpp4 functions during pronephrogenesis, suggesting mammalian kidney formation may be regulated by a similar mechanism demonstrated in this work. ENPP4 is highly expressed in human metanephros and kidney tumours and its expression increases in deceased donor kidney biopsies with delayed graft function after kidney transplantation (data from human protein atlas)[56]. Interestingly, *ENPP4* is localized close to *RUNX2* gene, whose mutations cause cleidocranial dysplasia (CCD)[57,58]. Furthermore, a child with CCD and crossed renal ectopia has been reported, and given our data, we can speculate that the ectopic kidney is attributable to *ENPP4* locus alterations[59].

We propose a potentially novel model of action of the lipidic pathway in kidney physiology, implicating either bioactive lipids distinct from LPA and S1P molecules or a novel non-catalytic interaction. The fact that a S1P receptor might be activated other than by S1P binding may explain the controversy regarding the beneficial actions of FTY720 in renal pathologies[60]. Moreover, our study raises potentially fascinating possibilities regarding regenerative therapies for renal diseases. As therapies for chronic renal failure are still lacking, the identification of a novel pathway enabling the generation of ectopic kidneys may provide useful insights to therapeutics that enhance human renal regeneration.

## Methods

**Ethics statement**. The work was carried out under a UK Home Office-approved animal procedures project license and approved by the University of Warwick Biological Ethics Committee.

**Enpp4 cloning and site-directed mutagenesis**. The *Xenopus* enpp4 cDNA (Accession number: BC 079717) was cloned into pcDNA3.1. Mouse *Enpp4* cDNA (Accession number: BC027749) was cloned into pCS2+ and pcDNA3.1. Site-directed mutagenesis of *Xenopus* enpp4 was performed using a PCR-based approach. For each mutant, 2 successive rounds of PCR were carried out using the Pfx polymerase (Invitrogen) following the manufacturer's protocol and using the primers listed below. The first round of PCR, performed using the *enpp4*-pRNA3 plasmid as template, allowed the amplification of two fragments of the enpp4 coding region, one upstream and containing the desired mutation (underscored in the primer sequence) and the other downstream and containing the mutation, respectively. For this, one amplification was performed using the upstream primer carrying out the mutation and the primer ORF downstream containing the stop codon and the other amplification using the downstream primer carrying out the mutation and the primer ORF upstream containing the ATG codon. The two PCR products were then mixed and a third PCR was carried out using this mixture as template using the upstream and downstream ORF primers, carrying out the *BamH*I and *EcoR*I restriction sites, respectively (in italic in the primer sequence). The final PCR product was digested by *BamH*I and *EcoR*I and inserted into the pCS2+ vector. The presence of the correct mutation was confirmed by sequencing for each mutant. The mutant enpp4 cDNA was then extracted from the pCS2+ and cloned into the pcDNA3.1. All constructs were verified by sequencing.

**mRNA synthesis and morpholino oligonucleotides**. Capped mRNAs were synthesized using mMESSAGE mMACHINE Kits (Ambion) from linearized plasmids. Plasmids used were *Xenopus* enpp4-pRNA3 (clone BC079717); mouse *Enpp4*-pCS2+; *Xenopus* mutant enpp4-pCS2+; *Xenopus* s1p5r.L-pCS2+ (clone DC111014); *Xenopus* s1p1r-pCMV-Sport6 (clone BC074356); *Xenopus* p2y10-pCMV-Sport6 (clone BC084356) and *Xenopus* rfng-pCMV-Sport6[28]. enpp4 MO1 (5′-atgaaaacccttccaaacatcttga-3′), enpp4 MO2 (5′-gaaatgtcacacacgcagctcctat-3′), enpp6 MO (5′-aacgtgctgtacttagccatgccac-3′), s1pr5.L MO (5′-catggtttcgtcaatcctt-tatttc-3′), s1pr5.S MO (5′-catggttcagtcaatgctttatctc-3′), rfng MO[28] and standard control MO (cMO) were designed and supplied by GeneTools, LLC.

**Embryo culture, dissection, microinjections and lineage staining**. *Xenopus* embryos were staged according to Faber and Nieuwkoop[61]. Kidney and pronephric anlagen dissections were performed in Barth X[33]. Each individual sample was injected into the lateral marginal zone of a ventral-vegetal blastomere (V2) at the

8-cell stage to target the pronephros. Pilot experiments were carried out to determine the *enpp4* and lipidic receptors mRNAs and MOs quantities to inject, based on their abilities to alter kidney development without affecting the overall morphology of the embryos. The *rfng* mRNA and MO dose was used as previously published[28]. The *LacZ* (250 pg) mRNA was used as a lineage tracer. *LacZ* mRNA was injected alone or in combination with standard MO as controls. Injected embryos were cultured to various developmental stages, fixed in MEMFA and stained for β-galactosidase activity (Red-Gal or X-Gal staining) to identify correctly targeted embryos. Only embryos that had normal pronephros formation on the uninjected side and correctly targeted β-galactosidase staining on the injected side were scored.

**Analysis of molecular marker expression in embryos**. Whole-mount immunohistochemistry was performed using 3G8 and 4A6 monoclonal antibodies as previously described[20]. Whole-mount in situ hybridization was carried out as previously described[62]. Anti-sense digoxigenin (DIG)-labelled RNA probes were synthesized from linearized template plasmids[24,28,63]. Either BM purple (Roche Applied Science) or NBT/BCIP (Roche Applied Science) or Fast Red/Naphtol AS/MX (Sigma) was used for the colour reaction. After bleaching, embryos were photographed with a magnification of ×10 for whole stage 41 embryos, ×20 for whole gastrula, neurula and early organogenesis stages embryos and ×32 for pronephric region.

**Acrylamide embedding, cryostat sectioning and Hoechst staining**. *X. laevis* embryos were embedded sectioned at 18 μm thickness and nuclear Hoechst staining performed[64].

**RT-PCR**. RT-PCR reactions were carried out on whole or dissected *X. laevis* embryos as described previously using the housekeeping gene *odc* as loading control. Quality of pronephric tissues dissections was assessed by amplification of the kidney markers *lhx1* and *pax8* and of the muscle marker *myf5*. Amplification conditions and primers sequences for the *enpp, lpar, s1pr, lhx1* and *pax8* genes have been previously published[16,32,65]. *Myf5* was amplified using the forward primer, 5′-actactacagtctcccaggacaga-3′ and the reverse primer, 5′-agagtctggaataggagggagca-3′, with the annealing temperature of 60 °C and 29 cycles. Each sample was analyzed in two independent embryo batches.

**Cell culturing and transient transfection**. Chinese hamster ovary (CHO) cells were cultured in HAMs F-12 (Gibco BRL) containing 10% foetal serum, NaHCO$_3$ at 1.176 g/l, 2 mM of glutamine, 5 U/ml penicillin and 37.8 U/ml streptomycin during the 48 h prior to transfection. Cells were then transfected for 24 h using the reagent Turbofect (Fermentas) with 1 μg of the eukaryotic constructs. The transfection medium was then removed and replaced with culture medium. Approximately 48 h post transfection, cells were fixed or harvested for analyses. As a control, CHO cells were transfected with empty vector pcDNA3.1.

**Anti-Enpp4 antibody production**. The anti-Xl Enpp4 polyclonal antibody was raised in rabbits by direct intramuscular injection of the *Xenopus laevis* wild-type enpp4-pcDNA3.1 plasmid followed by electroporation (Aldevron, LLC, USA). Rabbits were immunized three times, at day 0, day 28 and 56 and terminal bleed performed at day 70.

**Immunocytochemistry**. Immunocytochemistry was carried out on fixed unpermeabilized cells with polyclonal antibodies anti-XlEnpp4 used at 1/200 or anti-mEnpp4 (CR65; see ref. [31]) used at 1/400 and anti-rabbit IgG FITC (Sigma) at 1/80. The staining was recorded using a Nikon Optiphot/ Diginet camera system. Photographs were taken at a magnification of ×40.

**Electrophoresis and western blot**. Native membrane proteins were extracted from transfected cells using the ProteoExtract® Native Membrane Protein Extraction Kit (Calbiochem). Proteins from whole cells and from the membrane and soluble fractions were separated on a 12% SDS-PAGE protein gel. Part of the gel was stained with Coomassie blue and processed for mass spectrometry analysis (Pôle Protéomique, Plateforme Génomique Fonctionnelle de Bordeaux, Université de Bordeaux) and the other was transferred onto a PVDF membrane (Bio-Rad) overnight at 4 °C. To limit non-specific binding, the anti-XlEnpp4 antibody was pre-absorbed on untransfected CHO cells. The membrane was incubated overnight at 4 °C with 1:200 dilution of Enpp4 antiserum, washed and incubated in goat anti-rabbit IgG peroxidase secondary antibody (Sigma, dilution 1/2000) for 30 min at 20 °C. After several washes, immunoreactivity was detected by chemiluminescence (Western lighting Chemiluminescence Reagent Plus, Perkin Elmer).

**Lipid binding assay**. Hydrophobic membrane pre-spotted with bioactive lipids (ShingoStrips™ S-6000 and Membrane Lipids Strips™ S-6002; Echelon Biosciences) were blocked 1 h at 20 °C with 1% BSA in Tris-buffered saline 0.05% Tween-20 (TBST). All subsequent washes were performed in TBST. Blots were overlaid with proteins extracts from membrane fractions of CHO cells transfected with enpp4-pcDNA3.1 or

empty vector (dilution 1/30) in blocking buffer overnight at 4 °C. Membranes were washed and incubated with pre-absorbed Enpp4 serum (dilution 1/200) for 6 h at room temperature. After several washes, the membranes were incubated with goat anti-rabbit IgG peroxidase secondary antibody for 30 min at room temperature, washed and developed using enhanced chemiluminescence. To confirm the observed binding, nitrocellulose Hybond-C extra (GE Healthcare) membranes were spotted with 0 to 200 μM of PA (Sigma P-9511) or PS (Sigma P-6641) diluted into a mix of MeOH/CHCl$_3$/H$_2$O (2/1/0.8, v/v). Dried membranes were then treated as described above.

**Statistics and reproducibility**. All experiments were repeated several times, on different batches of embryos, and pronephric phenotypes were determined in a commonly used way, blind-coded, by comparing the injected and uninjected sides[28]. The percentages of the embryos displaying the discussed phenotypes are given in the text in bracket along with the total number of analysed embryos. All raw data and statistical analyses are presented in the Supplementary Tables (SI). Each histological analysis was numbered (see Supplementary Tables 1a, 2a, 3a and 5a) and statistically pairwise compared as indicated in the Supplementary Tables 1b, 2b, 3b and 5b. Chi-square statistical analysis could not be performed for all comparison of experiments since the assumption was not always fulfilled. Hence, 2×5 Fisher's exact test was therefore used. Bonferroni multiple testing correction was then applied to all statistical analyses. All the statistical analyses were performed using the R statistical software Core Team R[66].

**Reporting summary**. Further information on research design is available in the Nature Research Reporting Summary linked to this article.

## Data availability

The authors confirm that all the data supporting the findings of this study are available in this article and its Supplementary Information files. Unedited gels and western blot are also presented in Supplementary Figs. 8 and 9.

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

## Acknowledgements
We are grateful to Junichi Kyuno for his contribution to experimental design and data analysis. We thank E. Pera for the *raldh1a2*, *ralhd10* and *cyp26a1* in situ probes, A. Tocco for technical help and P. Jarrett for maintenance of frogs. We also thank J-W. Dupuy (Pôle Protéomique Plateforme Génomique Fonctionnelle Bordeaux, Université de Bordeaux) for MS analysis, and B. Arveiler for helpful comments regarding human pathologies. The work was supported by Wellcome trust grant 082071 as well as CNRS, University of Bordeaux and ANR.

## Author contributions
K.M. and E.A.J. designed research; K.M., E.A.J., S.B. and L.M.-P. performed research; K.M., E.A.J., C.P., E.B.-G. and L.M.-P. analysed data; K.M. and E.A.J. wrote the paper with contributions from all the authors.

## Competing interests
The authors declare no competing interests.
