## [Transparent Peer Review File · Communications Biology]

Reviewers' comments:

Reviewer #1 (Remarks to the Author):

I complement the authors on this extremely well crafted and carefully controlled study. This paper describes a novel function for enpp4 in determining the early renal specification and exposes a striking overexpression phenotype: duplicated or ectopic kidneys. Interestingly, the authors identify S1PR5 a receptor that enhances the activity of enpp4 and an interaction with phosphatidylserine. These results could be leveraged to improve renal organoid technologies and the like. Therefore, I believe this work is of great importance to the field.

In conclusion, I am deeply impressed by the depth the authors went through in following up on this extremely interesting phenotype. In general, the experiments are very well controlled. Knocked down experiments were validated with two independent MOs and convincing rescue experiments performed. Even the raw data and statistical analysis are provided for each figure. The microdissection studies are extremely impressive. All In situ hybridizations are informative, convincing and very interesting.

The authors have obviously performed many control experiments that provide ample evidence for their conclusions and I suggest to publish this paper with only very minor revisions.

Here are some minor suggestions on how to improve the manuscript:

Be more explicit in the intro about which pathologies lipidic and purinergic signaling is related to.

Please add results of statistical test to the description of the rescue experiment in the results section (Line 167-179), just like for most other data. The data is present in the supplements, but this inconsistency just caught my eye.

I am a bit confused that the authors generated an antibody against full-length Xenopus Enpp4, but then only used it to detect the overexpressed protein in COS cells, a feat easily achievable by fluorescence tagging. Were immunostainings of the endogenous protein in Xenopus performed? I don't find the data in Figure 5 to be very striking.

Correct typo (CCD not CDD) in line 404.

Line 409, consider changing the term "polemic" Def.: "a strong verbal or written attack on someone or something" – I doubt the authors mean to convey this notion.

I advise changing the reference to acute renal failure to "chronic" renal failure, Acute injury is less of a problem (tubular cells regenerate) than the chronic form (endpoint: dialysis / transplant).

I see Enpp4 in Suppl Table 4A not 4B, might A be the overexpressed sample?

Reviewer #2 (Remarks to the Author):

In their manuscript, Massé et al suggest that the regulation of purinergic and lipidic signalling pathways by enpp ectonucleotidases that have been observed in fully developed metanephric kidney, are also important for nephrogenesis. Their previous work in the Xenopus pronephros showed that various enpp ectonucleotidases are expressed in the developing embryonic kidney. One of these enpp genes was enpp4, which is expressed strongly in the developing pronephric tubules. In this paper, the authors perform functional experiments where they manipulate enpp4 expression and observe the

effects this has on pronephrogenesis. They find that *enpp4* overexpression causes ectopic anterior kidney formation and *enpp4* knockdown using a morpholino oligonucleotide reduces the size of the kidney. The authors then extend these functional findings of a role for *enpp4* in pronephrogenesis by experiments that suggest *enpp4* functions are regulated by the *s1pr5* receptor and interacts with phosphotyrosine.

The results are of interest and I am overall convinced that there is a role in pronephrogenesis for *enpp4*, mainly due to the striking ectopic phenotypes observed by its over-expression. I was less convinced by the authors description of the role *enpp4* has in regulating important signalling pathways such as RA/Notch/Wnt where I felt that more rigour for these experiments could have been sought by performing *enpp4* OE/KD experiments in the presence of agonist/antagonists of these pathways to observe the epistatic relationship of *enpp4* to these signalling pathways. However, I believe the results are of interest to developmental biologists and to the readership of Communications Biology. Below are a few comments that I would expect the authors to respond to before my accepting of this manuscript for publication.

Major comments

1. *lhx1* expression in Fig.1X; Do the authors have an explanation for why this gene isn't expanded at neurula stages whereas *pax8* is? Fig.1X also looks like *lhx1* might be reduced. This is peculiar and the manuscript would benefit from further discussion about why the authors think *lhx1* might be different to *pax8* in terms of its early expression profile. This is particularly important as the authors discuss (line 351 onwards) the loss of *lhx1* in Fig.2M, therefore *enpp4* seemingly is required for *lhx1* expression?
2. The rescue experiments are supportive that the morpholino is specific to *enpp4*, but the reduction in both the 3G8 and 4A6 region is peculiar given *enpp4* is not expressed in the distal region. Does *enpp4* knockdown with a morpholino perturb muscle formation and so is this the reason the whole pronephric field might be affected? The authors infer that a "smaller pronephros had formed" in response to the 3G8/4A6 staining in the morphants, hence why I ask. Of note however, the *clcnk* and *gata3* expression phenotypes lead me to interpret the data as it is the IT segment that is affected, not the DT or CT segments in the 4A6 domain. This would fit better with the *enpp4* expression profile, which in the authors previous paper looks to be in the PT and IT segments, but not DT or CT segments. If the authors agree, they should adjust their interpretation as written in the results section. It doesn't seem unlikely to me that if you lose the PT segments, then the IT segment (even though it is part of the 4A6 domain) would be affected and less convoluted as is observed in the wild-type pronephros (ie its size phenotype in the *enpp4* morphant could be indirect). I would also like to see a muscle marker in the morphants (similar to the *myh4* in situ show for the OE), just to confirm that no paraxial mesoderm defects are induced by *enpp4* loss or morpholino toxicity.
3. The manuscript lacks quantification, whilst some of the phenotypes are quite obvious (especially ectopic pronephrogenesis data), there are some more subtle phenotypes. The effect of *enpp4* mis-expression on RA/Notch/Wnt components in Fig.4 is a good example. The authors could take a random selection of their stained embryos and measure the sizes of the stained regions on the injected and contralateral control sides. If the authors no longer have the embryos stored, then repeating one of the injections and staining for at least one of the markers with measurements (in ImageJ or another imaging software) would suffice. It is difficult for the reader to gauge the phenotypes observed when no quantification other than n values is given. The authors state that "*cyp26a1* expression was normal in the pronephric region following *enpp4* mRNA and MO injection" on line 226, but it looks to be reduced on the injected side of the over-expressed embryo. Quantification would overcome any discrepancies in the phenotypes showed.

Minor comments

4. Line 165: *Ennp4*, should be *Enpp4*
5. Does the *enpp4* polyclonal antibody not work on cryosectioned *Xenopus* embryos?
6. In Figure 6K, the 3G8/4A6 domains are normal, but in Fig.6F they are reduced (as the authors describe on line 303/4). The authors presumably believe that the presence of *enpp4* in Fig.6K is the

reason why 3G8 staining is normal, but if s1pr5 knockdown is precluding enpp4 induction of ectopic 3G8 staining then it should also stop the normal domain of 3G8 expression also. Do the authors have an explanation for this result over and above what they say in lines 324/325?

Response to reviewers

We are pleased to submit a revised version of the manuscript COMMSBIO-21-0713-T entitled “The enpp4 ectonucleotidase regulates kidney patterning signalling networks in *Xenopus* embryos” for publication as an Article in *Communications Biology*.

We thank the editor and the expert reviewers for their constructive reviews, as well as for their suggestions to improve the quality of our manuscript. We are very grateful to the editor and two reviewers for their enthusiasm about the quality of the data and the interest of our work.

We have addressed all reviewers’ queries as detailed in a point-by-point response to their comments. We have revised the manuscript accordingly to specify and correct all points raised by reviewers. New data sets and quantification of expression alterations to the retinoic acid signalling pathway analysis have been performed and results are now precised in the manuscript. We also performed in response to reviewer #2 comment additional morphant experiments with a different muscle marker now included in revised supplemental Figure S2.

We hope that the manuscript meets now all the specification required to ensure its publication in *Communications biology*.

Point by point answer to reviewer’s comments

Reviewer #1 (Remarks to the Author):

I complement the authors on this extremely well crafted and carefully controlled study..... Therefore, I believe this work is of great importance to the field. In conclusion, I am deeply impressed by the depth the authors went through in following up on this extremely interesting phenotype....The authors have obviously performed many control experiments that provide ample evidence for their conclusions and I suggest to publish this paper with only very minor revisions.

Remarks:

1. *Be more explicit in the intro about which pathologies lipidic and purinergic signaling is related to*

Response: We modified the paragraph to specify pathologies and add a recent reference (reference 10). “In mammals, lipidic and purinergic pathways regulate metanephric physiology and their deregulation has been linked to acute renal injury and chronic kidney diseases including renal fibrosis polycystic kidney disease, renal cell carcinoma , nephritis or diabetic nephropathy^{6, 7, 8, 9, 10}”.

2. *Please add results of statistical test to the description of the rescue experiment in the results section (Line 167-179), just like for most other data. The data is present in the supplements, but this inconsistency just caught my eye.*

Response: p value is now indicated.

3. *I am a bit confused that the authors generated an antibody against full-length *Xenopus* Enpp4, but then only used it to detect the overexpressed protein in COS cells, a feat easily achievable by fluorescence tagging. Were immunostainings of the endogenous protein in *Xenopus* performed ? I don’t find the data in Figure 5 to be very striking.*

Response: the enpp4 polyclonal antibody was generated to detect and localize the expression of endogenous protein in *Xenopus* embryos. Antibodies recognized enpp4 in overexpressing system but despite intensive efforts, we failed to detect endogenous expression on cryosectioned *Xenopus* embryos. This is now stated in results section Line 268.

4. *Correct typo (CCD not CDD) in line 404.*

Response: this was corrected.

5 *Line 409, consider changing the term “polemic” Def.: “a strong verbal or written attack on someone or something” – I doubt the authors mean to convey this notion*

Response: we thank the reviewer and agree. The term was replaced by controversy.

6. *I advise changing the reference to acute renal failure to “chronic” renal failure, Acute injury is less of a problem (tubular cells regenerate) than the chronic form (endpoint: dialysis / transplant).*

Response: We replaced “acute” by “chronic”.

7. *I see Enpp4 in Suppl Table 4A not 4B, might A be the overexpressed sample?*

Response: The reviewer is correct. We modified the legend of Suppl Table 4.

Reviewer #2 (Remarks to the Author):

...I believe the results are of interest to developmental biologists and to the readership of *Communications Biology*. Below are a few comments that I would expect the authors to respond to before my accepting of this manuscript for publication.

1. *lhx1* expression in Fig.1X; Do the authors have an explanation for why this gene isn't expanded at neurula stages whereas *pax8* is? Fig.1X also looks like *lhx1* might be reduced. This is peculiar and the manuscript would benefit from further discussion about why the authors think *lhx1* might be different to *pax8* in terms of its early expression profile. This is particularly important as the authors discuss (line 351 onwards) the loss of *lhx1* in Fig.2M, therefore *enpp4* seemingly is required for *lhx1* expression?

Response: We thank the reviewer for this comment. It has been previously published that temporal expression induction of these two early pronephric genes is different (see reference Drews et al., 2011 below). Furthermore, Futel et al, 2015 demonstrated that early *pax8* and not *lhx1* expression is regulated by TRPP2—dependant Ca²⁺ signaling as loss of function of *pkd2* gene specifically targets *pax8* expression in kidney field. These 2 publications clearly demonstrate that *px8* and *lhx1* expression are regulated by distinct signaling pathways. We therefore modified our text (line 382) to discuss this point and added the 2 references.

2. The rescue experiments are supportive that the morpholino is specific to *enpp4*, but the reduction in both the 3G8 and 4A6 region is peculiar given *enpp4* is not expressed in the distal region. Does *enpp4* knockdown with a morpholino perturb muscle formation and so is this the reason the whole pronephric field might be affected?

Response: As we observed no alteration in somites at later stages, we considered very unlikely that muscle formation would be affected in *enpp4* loss of function. This assumption is supported by the data published by Futel et al., 2015 (reference 35).

3. The authors infer that a “smaller pronephros had formed” in response to the 3G8/4A6 staining in the morphants, hence why I ask. Of note however, the *clnk* and *gata3* expression phenotypes lead me to interpret the data as it is the IT segment that is affected, not the DT or CT segments in the 4A6 domain. This would fit better with the *enpp4* expression profile, which in the authors previous paper looks to be in the PT and IT segments, but not DT or CT segments. If the authors agree, they should adjust their interpretation as written in the results section. It doesn't seem unlikely to me that if you lose the PT segments, then the IT segment (even though it is part of the 4A6 domain) would be affected and less convoluted as is observed in the wild-type pronephros (ie its size phenotype in the *enpp4* morphant could be indirect).

Response: We agree with this comment. The text was modified to specify that “The *clnkb* expression in the intermediate tubules” (line 172) and in line 184-187 “These results suggest that *enpp4* knock-down affected pronephric pronephric tubule, especially proximal and intermediate segments, differentiation rather than just the proximal-distal patterning of pronephric tubule segmentation.

3.2. I would also like to see a muscle marker in the morphants (similar to the *myh4* in situ show for the OE), just to confirm that no paraxial mesoderm defects are induced by *enpp4* loss or morpholino toxicity.

Response: Somites phenotypes, using the same muscle marker as in the overexpression experiments (*myh4*), following *enpp4* loss of function were already indicated in the text (line 165) and in FigS2). However, as requested by the reviewer, we performed additional loss of function experiments (injection of *enpp4* MO1) and analysed the muscle phenotype with a distinct myogenic marker MyoD. As shown in the novel Fig. S2D' panel of revised Supplemental Figure S2, *myoD* expression is unchanged in somites, but *myoD* expression alteration was observed in the hypaxial muscle progenitors. Values were also added in Supplemental table 2A. New FigS2 and the added panel (yellow frame) is indicated below.

3.3. The manuscript lacks quantification, whilst some of the phenotypes are quite obvious (especially ectopic pronephrogenesis data), there are some more subtle phenotypes. The effect of *enpp4* mis-expression on RA/Notch/Wnt components in Fig.4 is a good example. The authors could take a random selection of their stained embryos and measure the sizes of the stained regions on the injected and contralateral control sides. If the authors no longer have the embryos stored, then repeating one of the injections and staining for at least one of the markers with measurements (in ImageJ or another imaging software) would suffice. It is difficult for the reader to gauge the phenotypes observed when no quantification other than *n* values is given. The authors state that “*cyp26a1* expression was normal in the pronephric region following *enpp4* mRNA and MO injection” on line 226, but it looks to be reduced on the injected side of the over-expressed embryo. Quantification would overcome any discrepancies in the phenotypes showed.

Response: We thank the reviewer for this comment. The text now specified whether these changes are significant or not. We performed new set of wholemount *in situ* hybridization experiments with *raldh1a2* or *cyp26a1* probes as previously illustrated (Fig. 4H et 4J) and measure the staining for *raldh1a2* and *cyp26a1* markers on injected and non-injected (NI) sides of embryos injected with *enpp4* MO1, MO2 or control MO (cMO) and with *enpp4* mRNA or LacZ mRNA alone. Comparison of surface stained for *raldh1a2* between non-injected and injected sides by paired Student t-test are not significantly different ($p > 0.05$ with MO1, $n=13$; MO2 $n=14$ and cMO $n=18$) (Panel A of the figure response 3.3 see below). In contrast, *enpp4* overexpression significantly increased the surface stained for *raldh1a2* compared to non-injected ($p < 0.01$, $n=10$) whereas LacZ mRNA injection has no effect ($p > 0.05$, $n=12$) (Panel B of the figure response 3.3). The number of particles stained for *cyp26a1* measured in absence or presence of MO1, ($n=11$) MO2 ($n=8$) or control MO (cMO) ($n=11$) or after *enpp4* overexpression ($n=12$) (Panel C-D of the figure response 3.3) are not significantly different ($p > 0.05$ in each condition) compared to the non-injected side. These quantified results are consistent with the statistical results based on the phenotypes indicated in supplemental Table 3A.

Figure for response 3.3: Summary of the quantification of the expression domain of *raldh1a2* and *cyp26a1*. A. Mean +/-sem of the surface domain of expression of *raldh1a2* in injected sides and non-injected (NI) after *enpp4* MO1, *enpp4*MO2 and cMO or (B) after *enpp4* mRNA injection. C. Mean +/-sem of the number of stained particles in the domain of expression of *Cyp26a1* in injected sides and non-injected (NI) after *enpp4* MO1, MO2 and cMO or (D) after *enpp4* mRNA injection.

Minor comments

4. Line 165: *Ennp4*, should be *Enpp4*

Response: this was corrected

5. Does the *enpp4* polyclonal antibody not work on cryosectioned *Xenopus* embryos?

Response: the *enpp4* polyclonal antibody was generated to detect and localize the expression of endogenous protein in *Xenopus* embryos. Antibodies recognized *enpp4* in overexpressing system but despite intensive efforts, we failed to detect endogenous expression on cryosectioned *Xenopus* embryos. This is now precised in results section Line 268.

6. In Figure 6K, the 3G8/4A6 domains are normal, but in Fig.6F they are reduced (as the authors describe on line 303/4). The authors presumably believe that the presence of *enpp4* in Fig.6K is the reason why 3G8 staining is normal, but if *s1pr5* knockdown is precluding *enpp4* induction of ectopic 3G8 staining then it should also stop the normal domain of 3G8 expression also. Do the authors have an explanation for this result over and above what they say in lines 324/325?

Response: The co-injection of *enpp4* mRNA and *s1pr5*.L MO does not result in normal 3G8 staining. The photography in Fig.6K was chosen to illustrate the reduction of ectopic pronephric tissues (size and number) indecid by *enpp4* overexpression. However, the raw data given in Table Sup 5A shows that more than 40% of the embryos display a reduced 3G8 staining and less than 50% of the co-injected embryos display a normal 3G8 staining. This is now indicated in the text line 351 'Moreover, the percentage of embryos injected with *enpp4* mRNA and *s1pr5*.L MO displaying a reduced 3G8 and 4A6 expression domain remains high (42,5% and 30% respectively, n = 40, Fig. 6K), most certainly due to the loss of function of *s1pr5*' .

REVIEWERS' COMMENTS:

Reviewer #1 (Remarks to the Author):

I find that all my suggestions were taken up by the authors and I now fully support publication of this important work in Communications Biology.

Reviewer #2 (Remarks to the Author):

The authors have given answers to all the questions I raised regarding the original manuscript. I therefore support accepting this manuscript for publication in Comms Bio.